# Effects of s-ketamine and midazolam on respiratory variability: A randomized controlled pilot trial

Oscar F. C. van den Bosch[1☯]*, Johan P. A. van Lennep[2☯], Ricardo Alvarez-Jimenez[1], Henriët van Middendorp[2], Andrea W. M. Evers[2], Monique A. H. Steegers[1], Patrick Schober[1], Stephan A. Loer[1]

1 Department of Anesthesiology, Amsterdam UMC location Vrije Universiteit, Amsterdam, the Netherlands,
2 Institute of Psychology, Leiden University, Leiden, the Netherlands

☯ Both authors contributed equally to this work.
* o.vandenbosch@amsterdamumc.nl

## Abstract

S-ketamine and midazolam are frequently used to provide sedation while maintaining spontaneous respiration. However, the effects of these agents on respiratory variability, which reflects the adaptability of the respiratory system, have not been thoroughly explored. We evaluated these effects in a randomized controlled pilot trial. This study was conducted as part of a randomized controlled trial originally designed to assess the effects of s-ketamine conditioning on pain sensitivity in patients with fibromyalgia syndrome. Participants were randomly assigned to receive an infusion of either s-ketamine ($0.3\,mg\,kg^{-1}\,h^{-1}$), midazolam ($0.05\,mg\,kg^{-1}\,h^{-1}$), or saline in a blinded fashion. Mean respiratory rate, variability of respiratory rate (VRR), and variability of tidal volume (VTV) were measured continuously and non-invasively with a bio-impedance method. Changes during drug infusion were compared in a linear mixed model to assess the effects of s-ketamine and midazolam compared to saline. Data were analyzed for 57 experiments in 28 participants. Their median baseline variabilities of respiratory rate and tidal volume were 0.19 (IQR: 0.16–0.25) and 0.23 (0.19–0.34), respectively. While mean respiratory rate was not affected, midazolam resulted in a significant decrease in both VRR (ß = −0.071, 95% CI: −0.120 to −0.021) and VTV (ß = −0.117, 95% CI: −0.170 to −0.062). In contrast, s-ketamine appeared to produce a smaller decrease in VTV (ß = −0.062, 95% CI: −0.118 to −0.003) with VRR remaining unaffected (ß = −0.036, 95% CI: −0.092 to 0.019). In conclusion, our study demonstrates that midazolam reduces respiratory variability, potentially impairing the adaptability of the respiratory system. In contrast, s-ketamine largely preserved respiratory variability, suggesting it may be a safer alternative for sedation in patients with impaired spontaneous breathing. Further studies are needed to assess the clinical implications of these observations in patients undergoing sedation.

**Data availability statement:** All data files are publicly available in anonymized form

in the DataverseNL repository (https://doi.org/10.34894/BAZOQF).

**Funding:** The Dutch Arthritis Society (ReumaNL) and the NWO Stevin grant, both awarded to A. Evers. The funders had no role in study design, data collection and analysis, decision to publish, or preparation of the manuscript.

**Competing interests:** The authors have declared that no competing interests exist.

## Introduction

Respiratory variability, defined as the extent of fluctuations in breathing parameters, is an essential physiological phenomenon influenced by both endogenous and exogenous factors [1]. It reflects the complex interplay between central respiratory control, peripheral feedback mechanisms, and autonomic regulation. In general, it is considered an indicator of respiratory function, and decreased variability may indicate poor adaptability and impending respiratory failure in critical care settings [2].

In anesthesiology, emergency medicine, and critical care medicine, two frequently used anesthetic drugs are s-ketamine and midazolam. They modulate the central respiratory regulation in different ways. Midazolam, a benzodiazepine, exerts its effects primarily through the potentiation of GABA activity. It is widely used in procedural sedation and anesthesia induction. Midazolam depresses respiratory function by reducing respiratory rate and tidal volume [3]. In contrast, s-ketamine acts as an NMDA-receptor antagonist and is a dissociative anesthetic associated with minimal respiratory depression, which can even stimulate respiration under certain conditions [4]. Yet, its impact on respiratory variability remains incompletely understood. During procedural sedation with propofol and remifentanil, midazolam reduces respiratory variability while s-ketamine preserves variability [5]. However, little is known about their singular effects on respiratory function.

Therefore, this pilot study compares respiratory variability in participants undergoing infusion of s-ketamine, midazolam, or saline during restful waiting. Specifically, we sought to assess the effects of these interventions on mean respiratory rate, variability of respiratory rate, and variability of tidal volume. This study was conducted as part of a larger randomized controlled trial primarily aimed at assessing the effects of s-ketamine on pain sensitivity in patients with fibromyalgia syndrome.

## Methods

### Trial design and ethics

This was a randomized controlled trial comparing s-ketamine, midazolam, and saline, conducted from February 7, 2023 until March 11, 2024, at the Amsterdam University Medical Center, location VUmc, Amsterdam, the Netherlands. The primary aim of the study was to investigate whether pharmacological conditioning with s-ketamine, compared to conditioning with placebo treatment, reduces pain hypersensitivity in patients with fibromyalgia. This report addresses the pre-planned secondary aim: to investigate the effects of s-ketamine and midazolam on respiratory variability. In the absence of prior data in this specific context, this study was designed as a pilot with limited sample size, not powered for definitive hypothesis testing regarding respiratory variability. The study protocol was approved by the Medical Ethics Committee of Leiden University (The Netherlands) on 14 September 2022 (reference NL73444.058.21) and was prospectively registered in the EudraCT database on 27 May 2022 (reference 2019-004812-73). All participants provided written informed consent. The reporting of this study adhered to the CONSORT 2025 statement. The authors used ChatGPT 4.0 to refine the clarity of the content. The authors reviewed and edited the content as needed and take full responsibility for this work.

## Participants

Women (18–75 years) diagnosed with fibromyalgia by a rheumatologist and able to understand and speak Dutch were eligible to participate in this study. The exclusion criteria were (i) pulmonary obstructive or restrictive disease, (ii) neuro-muscular disease, (iii) hypertension or any other severe cardiovascular comorbidity, (iv) a medical diagnosis other than fibromyalgia explaining the chronic pain symptoms, (v) presence of any severe psychiatric comorbidity not related to symptoms of fibromyalgia, (vi) allergy for s-ketamine, midazolam, ondansetron, or flumazenil, (vii) previous experience with s-ketamine in a medical setting or recreational use, (viii) current or previous dependence on strong analgesics, alcohol, or drugs, (ix) caffeine use within 12 h prior to the study visit, (x) a body mass index > 35 kg/m$^2$, and (xi) pregnancy or breastfeeding.

Potential subjects received detailed information, including potential side effects of the pharmacological treatment. The main experimenter verified eligibility to participate, and a consultant anesthesiologist determined if the patient could receive study medication. Eligible participants provided written informed consent. The clinical pharmacy of the Amsterdam UMC, location VUmc, randomized participants in a 1:1:1 ratio in blocks of variable sizes with CastorEDC, and subsequently prepared the blinded study intervention. The experimenters and participants were blinded to the allocation of pharmacological intervention.

## Procedures

This study consisted of three visits per participant at an interval of one visit per week. Every visit consisted of the following elements: preparations, baseline recordings, administration of intervention, and intervention recordings. Participants received the same pharmacological intervention at each visit.

During preparations and baseline recordings, participants laid down on a bed, an intravenous catheter was placed, and monitoring was commenced, including non-invasive blood pressure and pulse oximetry measurements. Respiratory measurements were performed non-invasively with an adhesive chest electrode using the bio-impedance technique to continuously measure and record respiratory rate and changes in tidal volume (Respiratory Motion ExSpiron 1xi). Respiratory parameters were averaged over 60-second intervals and saved for further analysis. Participants filled out the Fibromyalgia Impact Questionnaire (FIQR). Participants were asked to provide their chronic pain intensity on a numeric rating scale (NRS) ranging from 0 (no pain) to 10 (worst pain imaginable).

Following baseline recordings, participants were instructed about the administration of study medication over the next hour. Participants received either s-ketamine, midazolam, or saline treatment in a double-blinded fashion. S-ketamine was administered in a step-up fashion at a dose of 0.1 mg kg$^{-1}$ h$^{-1}$ during 20 mins, followed by 0.2 mg kg$^{-1}$ h$^{-1}$ during 20 mins, followed by 0.3 mg kg$^{-1}$ h$^{-1}$ during 20 mins. This dosage was determined based on previous studies evaluating the therapeutic efficacy of S-ketamine in the treatment of fibromyalgia syndrome, as well as recommendations from international guidelines [6,7]. Midazolam was administered in a similar fashion at a dose of 0.017 mg kg$^{-1}$ h$^{-1}$ during 20 mins, followed by 0.033 mg kg$^{-1}$ h$^{-1}$ during 20 mins, and then 0.050 mg kg$^{-1}$ h$^{-1}$ during 20 mins. This dosage was determined based on the goal to induce conscious sedation [8]. Saline was administered in a similar stepwise manner to maintain appropriate blinding.

Respiratory variability was calculated over 20 minute intervals, similar to prior studies [5,9], and aligned with the 20 minute infusion period at the maximum study drug dose. Baseline respiratory recordings were obtained during a restful 20-minute period that began 30 minutes before the start of drug infusion. Intervention recordings were collected during a 20-minute period starting 45 minutes after initiation of drug infusion, corresponding to the time when participants were receiving the maximum dose of the study medication.

## Outcomes

The three co-primary outcomes were changes in (1) mean respiratory rate, (2) variability of respiratory rate, and (3) variability of tidal volume, measured as the difference between baseline recordings taken *before* drug administration, and intervention recordings taken *during* drug administration. [5,9]

## Statistical analysis

Data were analyzed using R (R Core Team, Vienna). Respiratory variability was calculated using a quantitative time-series evaluation. Variabilities in respiratory parameters were calculated using the coefficient of variation, which is defined as the ratio of the standard deviation to the mean of a time series [2]. The statistical association between the change in respiratory variability and group allocation was assessed using linear mixed-effects models to account for repeated measurements (i.e., three visits), using the `lme4` and `lmertest` packages [10]. The change in respiratory parameters from the baseline period to the intervention period was used as the dependent (outcome) variable. Group allocation (i.e., s-ketamine, midazolam, or saline) was used as a fixed effect, with saline as the reference condition, and participant identifier was used as a random effect (random intercept). In a post hoc analysis, midazolam was used as the reference condition to assess differences between the effects of s-ketamine and midazolam. No interaction terms were added to avoid overfitting. Bootstrapping with 5,000 iterations was applied to generate empirical confidence intervals, reducing reliance on parametric assumptions and improving robustness in small-sample inference. Effect sizes with 95% confidence intervals were calculated as standardized correlation coefficients. Significance was defined as a p-value of < 0.05. This study's sample size was determined based on the primary objective of evaluating pharmacological conditioning effects on fibromyalgia treatment. As such, it was not specifically powered for analyses related to respiratory variability.

## Results

From February 7, 2023 until March 11, 2024, 54 potential participants were recruited, and 28 participants were included and randomized following written informed consent (Fig 1). The first, second, and third visits were completed by 28 (100%), 24 (86%), and 21 (75%) participants, respectively. Complete respiratory datasets were available for 57 of 73 (78%) of visits. Data were incomplete for other visits due to (i) missing baseline measurements, (ii) inadequate capture of respiratory signals, and (iii) inadequate storage of the recordings in the respiratory monitor. The included participants had a median age of 54 (IQR: 49–61) years and a median body mass index of 27.7 (25.7 to 30.4) kg/m$^2$. Regarding baseline respiratory recordings, the median respiratory rate, median variability of respiratory rate, and median variability of tidal volume were 14.6 (13.3 to 16.6) breaths/min, 0.19 (0.16 to 0.25), and 0.23 (0.19 to 0.34), respectively. Details on participant characteristics and baseline respiratory recordings according to group allocation are shown in Table 1. Participants with

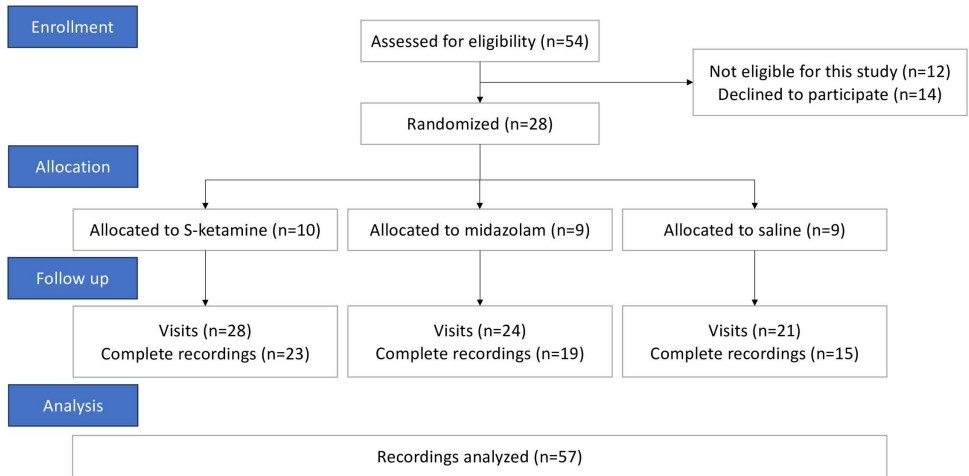

**Fig 1. Flow chart of the study participants.** Shown are enrollment, allocation, follow-up, and analysis of patients. The study compared respiratory variability during infusion of s-ketamine, midazolam, or saline.

**Table 1. Baseline patient characteristics.**

|  | S-Ketamine | Midazolam | Saline |
|---|---|---|---|
|  | N = 10/ 23* | N = 9/ 19* | N = 9/ 15* |
| **Age, years** | 48 (40–58) | 53 (48–58) | 56 (41–69) |
| **Weight, kg** | 74 (69–89) | 80 (61–84) | 79 (68–83) |
| **BMI, kg m⁻²** | 30 (26–31) | 26 (21–30) | 27 (23–30) |
| *Baseline respiratory parameters* |  |  |  |
| **Mean respiratory rate,/min** | 15.4 (13.9 to 16.7) | 15.0 (13.2 to 17.4) | 13.7 (13.0 to 14.5) |
| **Variability of respiratory rate, cv** | 0.21 (0.16 to 0.27) | 0.19 (0.13 to 0.24) | 0.20 (0.18 to 0.24) |
| **Variability of tidal volume, cv** | 0.25 (0.20 to 0.37) | 0.25 (0.19 to 0.35) | 0.20 (0.16 to 0.25) |

Data are median (IQR). cv coefficient of variation.

*number of participants/ number of visits.

incomplete data had a higher weight compared to those with complete data (84 [77–93] vs 73 [63–84] kg, p = 0.025), while other participant characteristics showed no significant differences.

The effects of s-ketamine and midazolam on respiratory parameters are shown in Fig 2. Administration of s-ketamine or midazolam did not significantly change the mean respiratory rate. Administration of midazolam, but not s-ketamine, was associated with a 37% decrease in variability of respiratory rate; with a corresponding effect size of −0.071 (95% CI: −0.121 to −0.019). Administration of midazolam was associated with a 51% decrease in variability of tidal volume; with a corresponding effect size of −0.117 (95% confidence interval [CI]: −0.170 to −0.062). Administering s-ketamine was associated with a smaller but significant decrease in variability of tidal volume of 27%; effect size −0.062 (95% CI: −0.119 to −0.003).

Post hoc analysis showed significant differences between s-ketamine and midazolam in all three respiratory parameters. All calculated effect sizes are shown in S1 Table.

## Discussion

This randomized controlled pilot trial examined the effects of s-ketamine and midazolam on respiratory variability. While no effects were found on mean respiratory rate, midazolam significantly attenuated the variability of respiratory rate, and to an even greater extent, the variability of tidal volume. S-ketamine decreased tidal volume variability but not respiratory rate variability.

The reduction in the variability of both respiratory rate (−37%) and tidal volume (−51%) caused by midazolam is clinically significant and warrants further attention. Interestingly, the magnitude of this decrease in respiratory rate and tidal volume is similar to the differences seen in the intensive care unit between patients who fail and succeed at extubating attempts following prolonged ventilation [11–14]. However, we acknowledge that spontaneous breathing trials are typically conducted in the absence of sedatives, whereas our study involved active administration of a sedative agent. While the similarity in magnitude is noteworthy, this should be interpreted with caution and should only provide some clinical context rather than imply equivalence. Also, a similar reduction was found in a study comparing respiratory variability during wakefulness versus non-pharmacological sleep in young participants [15].

In comparison to the effects of midazolam, s-ketamine produced a smaller decrease in tidal volume variability while respiratory rate variability remained unaffected. This finding suggests that s-ketamine largely preserves normal respiratory variability. The maintenance of respiratory variability under s-ketamine is consistent with its well-documented ability to support spontaneous ventilation while providing sedation and analgesia [16]. Also, s-ketamine stimulates noradrenergic neurons, prolongs synaptic action, and inhibits catecholamine uptake, thereby provoking a hyperadrenergic state that subsequently stimulates respiration [17,18].

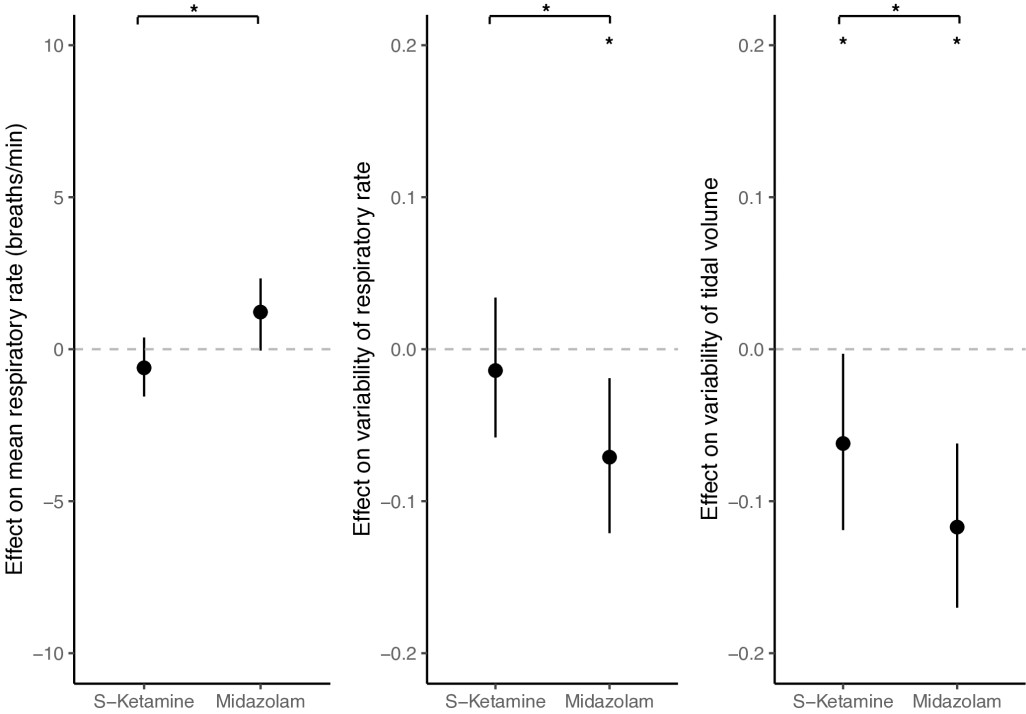

**Fig 2. Effects of s-ketamine and midazolam on respiratory parameters.** Effect sizes are shown as standardized beta coefficients (with 95% confidence intervals) from a linear mixed model with the difference in respiratory parameter before vs during study medication as the outcome variable, compared to saline. Variability is calculated as the coefficient of variation. * denotes statistical significance ($p < 0.05$). All calculated effect sizes are shown in S1 Table.

Regarding the mechanistic pathway, we suggest that midazolam suppresses fluctuations in breathing by enhancing inhibitory neurotransmission (i.e., GABA agonism) in the brainstem respiratory centers. Benzodiazepines reduce the responsiveness of the respiratory centers, decrease autonomic influences on respiration, and diminish cortical influence. Additionally, midazolam may impair upper airway dilator muscles and may increase upper airway resistance [19]. The resulting limitation in airflow may explain the significant reduction in tidal volume variability, as breathing becomes shallower and more uniform. Furthermore, reduced consciousness makes chemical control of respiration less stable, despite a concurrent decrease in chemoresponsiveness to hypercapnia [1,20]. In contrast to midazolam, s-ketamine abolishes the coupling between loss of consciousness and upper airway dilator muscle dysfunction [21]. This may explain why the effects on tidal volume variability are more pronounced under midazolam compared to s-ketamine. The preservation of respiratory variability under s-ketamine is likely due to its specific pharmacological effects involving NMDA receptor antagonism and interactions with monoaminergic pathways, which together support respiratory control [3].

The differential effects of s-ketamine and midazolam on respiratory variability may have important clinical implications in the setting of anesthesiology and critical care medicine, as they are frequently used as sedatives for both spontaneously breathing and mechanically ventilated patients. We observed that midazolam affects the adaptability of the respiratory system even at doses that do not alter the mean respiratory rate. This could suggest that midazolam may impair the ability of the respiratory system to adjust to changing physiological demands. These effects could be harmful in patients who are already at risk of failing to adapt to dynamic clinical conditions, such as those with chronic respiratory disease, sepsis, or neurological impairment. Our results contribute to a broader understanding of respiratory variability as a marker of autonomic and respiratory control. Reduced variability may indicate suppression of physiological reflexes, whereas

preserved variability suggests maintained adaptability. Given these distinctions, the selection of sedatives should consider not only the depth of sedation but also its impact on respiratory control, particularly in patients with underlying respiratory or neurological disorders.

In the setting of procedural sedation, midazolam's pronounced reduction in variability may be useful in settings requiring stable respiration, such as endobronchial procedures, but it also necessitates careful monitoring due to the risk of respiratory depression. In contrast, the preservation of respiratory variability with s-ketamine suggests it is a safer option for patients who are at risk of respiratory compromise during procedural sedation.

We would like to acknowledge the following limitations. Our pilot study was performed in patients with fibromyalgia syndrome who were otherwise healthy. Nevertheless, we believe that similar effects may be observed in other populations. This is particularly of relevance in patients with compromised respiratory function. Also, there is a paucity of studies and a lack of established clinical practices regarding non-invasive monitoring of respiratory variability. Respiratory data were incomplete for 22% of study visits. However, the use of a linear mixed model allowed for unbiased estimation of effects and was the most appropriate analytical approach given the nature and assumed randomness of the missing data. Additionally, our patients received only s-ketamine, midazolam, or saline, whereas sedation in clinical practice often involves combinations of multiple agents. Dosing strategies may also differ in routine clinical practice. Future studies should investigate with larger sample size how these medications interact with other anesthetic or sedative agents commonly used in critical care and perioperative settings.

## Conclusions

In conclusion, our findings suggest that midazolam significantly reduces the variability of respiratory rate and variability of tidal volume, leading to a more regular and less adaptable breathing pattern. This could be concerning in settings where respiratory adaptability is crucial for patients who are already at risk of respiratory failure. In contrast, s-ketamine preserves respiratory variability, which may offer advantages in critically ill patients who require sedation but need to maintain spontaneous breathing. Further research is needed to evaluate these effects in critically ill populations and to better understand the implications for sedative management in anesthesiology and intensive care medicine.

## Supporting information

**S1 Table. Effects of s-ketamine and midazolam on respiratory parameters: complete effect sizes in a linear mixed model.**
(DOCX)

**S1 File. CONSORT Checklist 2025.**
(DOCX)

**S2 File. Study protocol English.**
(PDF)

## Author contributions

**Conceptualization:** Oscar F. C. van den Bosch, Johan P. A. van Lennep, Ricardo Alvarez-Jimenez, Henriët van Middendorp, Andrea W. M. Evers, Monique A. H. Steegers, Stephan A. Loer.

**Data curation:** Johan P. A. van Lennep.

**Formal analysis:** Oscar F. C. van den Bosch, Johan P. A. van Lennep.

**Investigation:** Oscar F. C. van den Bosch, Ricardo Alvarez-Jimenez.

**Methodology:** Oscar F. C. van den Bosch, Ricardo Alvarez-Jimenez, Henriët van Middendorp, Patrick Schober, Stephan A. Loer.

**Project administration:** Henriët van Middendorp, Andrea W. M. Evers, Monique A. H. Steegers, Stephan A. Loer.

**Resources:** Henriët van Middendorp, Andrea W. M. Evers, Monique A. H. Steegers, Stephan A. Loer.

**Software:** Oscar F. C. van den Bosch, Patrick Schober.

**Supervision:** Henriët van Middendorp, Andrea W. M. Evers, Monique A. H. Steegers, Patrick Schober, Stephan A. Loer.

**Validation:** Patrick Schober.

**Visualization:** Oscar F. C. van den Bosch.

**Writing – original draft:** Oscar F. C. van den Bosch.

**Writing – review & editing:** Oscar F. C. van den Bosch, Johan P. A. van Lennep, Ricardo Alvarez-Jimenez, Henriët van Middendorp, Andrea W. M. Evers, Monique A. H. Steegers, Patrick Schober, Stephan A. Loer.

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
