## [Decision Letter · Decision Letter 0]

27 May 2025

PONE-D-25-20310Effects of S-Ketamine and Midazolam on Respiratory Variability A Randomized Controlled Pilot TrialPLOS ONE

Dear Dr. van den Bosch,

Thank you for submitting your manuscript to PLOS ONE. After careful consideration, we feel that it has merit but does not fully meet PLOS ONE’s publication criteria as it currently stands. Therefore, we invite you to submit a revised version of the manuscript that addresses the points raised during the review process.

We look forward to receiving your revised manuscript.

Kind regards,

Kartikeya Rajdev, MD

Academic Editor

PLOS ONE

Journal Requirements:

3. In the online submission form, you indicated that because the data contains potentially identifying patient information, they are available from the corresponding author for researchers who meet the criteria for access to confidential data. The minimal anonymized dataset can by provided by the corresponding author.

5. Please remove all personal information, ensure that the data shared are in accordance with participant consent, and re-upload a fully anonymized data set.

Reviewers' comments:

Reviewer's Responses to Questions

**Comments to the Author**

1. Is the manuscript technically sound, and do the data support the conclusions?

Reviewer #1: Partly

Reviewer #2: Yes

Reviewer #3: Partly

Reviewer #4: Partly

Reviewer #5: Yes

Reviewer #6: No

2. Has the statistical analysis been performed appropriately and rigorously? 

Reviewer #1: Yes

Reviewer #2: Yes

Reviewer #3: Yes

Reviewer #4: Yes

Reviewer #5: Yes

Reviewer #6: Yes

3. Have the authors made all data underlying the findings in their manuscript fully available?

Reviewer #1: Yes

Reviewer #2: No

Reviewer #3: No

Reviewer #4: No

Reviewer #5: Yes

Reviewer #6: Yes

4. Is the manuscript presented in an intelligible fashion and written in standard English?

Reviewer #1: Yes

Reviewer #2: Yes

Reviewer #3: Yes

Reviewer #4: Yes

Reviewer #5: Yes

Reviewer #6: Yes

5. Review Comments to the Author

Reviewer #1: The authors provide an appropriate background that clearly introduces the concept of respiratory variability, its clinical importance, and the current knowledge gaps in the field. This context is helpful in establishing the relevance and need for the study. The manuscript is appropriately referenced, citing key studies that support the rationale for the investigation.

However, a few areas warrant clarification and improvement:

Exclusion Criteria:

The exclusion criteria do not mention respiratory diseases, which is a significant omission. Given that such conditions can independently affect respiratory variability, it would be important to state whether participants with respiratory disorders were excluded.

Terminology Consistency:

In line 89, the term “breathing variability” is used, while “respiratory variability” is used elsewhere. For consistency and clarity, the manuscript should consistently use “respiratory variability.”

Methods – Procedures Section:

The description of procedures lacks clarity and sufficient detail. Since the study specifically evaluates respiratory variability in response to ketamine and midazolam infusions, the inclusion of the 4th visit for saline only administration details appears unnecessary and may confuse readers. Consider streamlining this section by removing unrelated procedural details.

Dose Rationale:

The rationale behind the chosen doses and duration of ketamine and midazolam administration should be clearly explained. Are these doses based on an established protocol for fibromyalgia treatment?

Outcomes Section:

Lines 150–154, which describe the timing of respiratory recordings, would be more appropriately placed under the “Procedures” subsection. Additionally, clarification is needed regarding the decision to obtain measurements only during a 20-minute window starting 45 minutes after infusion initiation. Why were measurements not collected throughout the infusion period?

Statistical Methods:

It appears that a quantitative time-series evaluation was used to assess respiratory variability. If so, this method should be explicitly stated under the “Statistical Analysis” section.

Missing Data:

In line 178, the manuscript notes that respiratory data were only available for 78% of visits. The authors should explain the reasons for the 22% data loss, especially since the study was conducted in a controlled environment.

Baseline Characteristics:

There are notable differences in baseline characteristics—e.g., median age (48 vs. 53) and weight (74 vs. 80)—between the ketamine and midazolam groups. Was a statistical comparison performed to determine whether these differences were significant and if they may have influenced the results?

Table Content:

The table includes fibromyalgia impact scores. It would be helpful to clarify the relevance of this score to the study’s outcomes. Additionally, data from the saline group are presented in the table, but no corresponding respiratory variability data or analysis is provided or discussed. If this data is not directly relevant, consider removing it to maintain focus.

Discussion Clarity:

In lines 222–225, the authors compare their findings to differences observed in ICU patients undergoing extubation trials. However, spontaneous breathing trials are typically performed when sedatives are withheld. The relevance of this comparison to the current study—where sedatives were actively administered—should be clarified and more appropriately discussed.

Overall, the manuscript explores an important and novel area, but addressing the points above will enhance its clarity, scientific rigor, and clinical applicability.

Reviewer #2: Overall, well written manuscript. I would recommend adding a few more details (as below) if seems appropriate for the audience to better understand the study design.

1). Did the saline group experience increase in rate of their saline infusion to better align with the increase in dosing for both Midazolam and S-ketamine (to minimize bias)

2) Please add a clarification sentence that the same group of patients received the same medications (S-ketamine, midazolam or saline) all 3 times.

Reviewer #3: I commend the authors for conducting a well-protocolized study on the respiratory depression effects of midazolam compared to S-ketamine. The findings suggest that S-ketamine may be a safer option for protecting the respiratory system's adaptability.

Although suppression of respiratory drive is a well-recognized effect of benzodiazepines via GABA-ergic stimulation, addressing a few remaining questions would enhance the study's clarity and impact. Some revisions could help present the findings more accurately and strengthen the overall conclusions.

A. The noninvasive bioimpedance technique (ExSpiron®Xi) has shown validation in some smaller trials compared to spirometry. However, the respiratory rate (VRR) and tidal volume (VTV) derived from bioimpedance need wider acceptance in clinical practice. While the bioimpedance-derived respiratory rate and tidal volume correlate well with spirometry, minute ventilation does not. This study focuses on non-invasive monitoring of respiratory variability, and we are uncertain if these findings are reproducible in similar studies or in clinical practice.

B. Respiratory data were only available in 78% of the visits. The remaining 22% missing data could have significantly altered the study’s findings.

C. In the study, the maximum dosing was 0.3 mg/kg/h for S-ketamine and 0.05 mg/kg/h for midazolam. The authors did not provide participants' median body weight, but the median BMI was 27.7. For an 80 kg participant, this translates to a maximum dose of 24 mg/h of S-ketamine and 4 mg/h of midazolam. The dosing regimen in clinical practice may significantly differ, potentially affecting the respiratory depressive effects of these medications.

D. In the discussion section (page 18, line 222), the author states, “Interestingly, the magnitude of this decrease in respiratory rate and tidal volume is similar to the differences seen in the intensive care unit between patients who fail and succeed at extubating attempts following prolonged ventilation.” However, I believe this interpretation stretches the study's findings. The research focused on fibromyalgia patients who were otherwise healthy and not on other sedatives/analgesics. In the ICU, many intubated patients have respiratory issues and may be receiving fentanyl or other analgesics along with sedatives. Thus, the VRR or VTV observed here might not apply to ICU patients.

Reviewer #4: This article presents results from a pilot study comparing ketamine to midazolam in terms of respiratory effects.

There are several points which need clarification.

1. The background seems very brief and it is difficult to understand exactly what is novel about the work to be undertaken.

2. Did each subject get each treatment? Or could a subject get the same treatment at each visit? Is this a crossover study? Or - each subject got the same treatment 3 times? Please describe more clearly. (I understand now they got the same treatment 3 times, but this is not clear as written.)

3. No sample size calculation is presented, but it is presented in the protocol. Please present a sample size calculation here.

4. What about this study makes it a pilot? Please justify using this descriptor.

5. How was missing data dealt with?

6. Were any subgroup analyses planned? Conducted?

7. The objective as written in the background only mentions respiratory variability, while rate, variability and variability of tidal volume are used as outcomes. In the outcomes it is described as "respiratory effects".

8. Please indicate in the figure 2 whether ketamine is different from medazolam.

9. Figure 2 - did these results account for the repeated measures nature of the data? If not, please revise.

10. Please give a more thoughtful consideration of missing data. Were the participants who dropped out different from those who completed all visits? How?

11. Lines 229 in the discussion - please present these results in the results section as well. (See comment #8)

12. line 254 - unclear from results presented if the effects are different.

13. Please also mention the small sample size and high rate of missingness as limitations - what effect may this have on results?

Reviewer #5: Extremely good study signifying the preference for Ketamine to prevent respiratory depression.

Well know use of ketamine for conscious sedation for emergency procedures is justified with objective lung parameters that clearly shows that its a better choice in respiratory patients.

Preference of ketamine is justified at molecular level.

Reviewer #6: This study has only 28 female patients, and needs a larger sample size.

Following Corrections/ additions are my suggestions: S-ketamine is not used commonly in anesthesia, and Midazolam is also used as an anxiolytic.

Exclusion criteria: should include patients with COPD, severe OSA, on home oxygen, heavy smokers, and allergy to benzodiazepines and ketamine.

They need to add details of questionnaires.

They should have monitored Co2, minute volume and VBGs.

Paragraph one in Page 5 and 6 and outcomes section need better flow of English.

Why did they choose this speicfic ketamine and Midazolam infusion dose?

Any info about quality and side-effects of sedation medications?

6. PLOS authors have the option to publish the peer review history of their article (what does this mean?). If published, this will include your full peer review and any attached files.

Reviewer #1: **Yes: **Sathish Kumar Krishnan

Reviewer #2: No

Reviewer #3: **Yes: **Abhisekh Sinha Ray

Reviewer #4: No

Reviewer #5: **Yes: **Karan Puri

Reviewer #6: No

---

## [Author Response · Author response to Decision Letter 1]

17 Jun 2025

Please find attached cover letter.

---

## [Decision Letter · Decision Letter 1]

15 Aug 2025

Effects of s-ketamine and midazolam on respiratory variability: A randomized controlled pilot trial

PONE-D-25-20310R1

Dear Dr. van den Bosch,

We’re pleased to inform you that your manuscript has been judged scientifically suitable for publication and will be formally accepted for publication once it meets all outstanding technical requirements.

Kind regards,

Kartikeya Rajdev, MD

Academic Editor

PLOS ONE

Additional Editor Comments (optional):

The submission was reviewed by six experts in the field. After carefully considering the revised manuscript, I believe the concerns raised during the review process have been acknowledged, including those outlined by Reviewer 6, who recommended rejection. Specifically, the authors acknowledged the limitations, including the small sample size, and discussed them transparently in the revised manuscript.

While Reviewer 6 raised valid points, I believe that you have responded appropriately to those concerns. In light of the revisions and the overall strength of the reviews, I concluded that the manuscript is acceptable for publication in its current form.

Thank you for choosing PLOSOne.

Kartikeya Rajdev, MD

Reviewers' comments:

Reviewer's Responses to Questions

**Comments to the Author**

1. If the authors have adequately addressed your comments raised in a previous round of review and you feel that this manuscript is now acceptable for publication, you may indicate that here to bypass the “Comments to the Author” section, enter your conflict of interest statement in the “Confidential to Editor” section, and submit your "Accept" recommendation.

Reviewer #1: All comments have been addressed

Reviewer #2: All comments have been addressed

Reviewer #3: All comments have been addressed

Reviewer #4: All comments have been addressed

Reviewer #6: (No Response)

2. Is the manuscript technically sound, and do the data support the conclusions?

Reviewer #1: Yes

Reviewer #2: Yes

Reviewer #3: Partly

Reviewer #4: Yes

Reviewer #6: No

3. Has the statistical analysis been performed appropriately and rigorously? 

Reviewer #1: Yes

Reviewer #2: Yes

Reviewer #3: Yes

Reviewer #4: Yes

Reviewer #6: Yes

4. Have the authors made all data underlying the findings in their manuscript fully available?

Reviewer #1: Yes

Reviewer #2: Yes

Reviewer #3: Yes

Reviewer #4: Yes

Reviewer #6: No

5. Is the manuscript presented in an intelligible fashion and written in standard English?

Reviewer #1: Yes

Reviewer #2: Yes

Reviewer #3: Yes

Reviewer #4: Yes

Reviewer #6: Yes

6. Review Comments to the Author

Reviewer #1: The authors have revised the manuscript based on reviewers' feedback and have provided appropriate explanations where appropriate. The manuscript can now be considered for publication.

Reviewer #2: This updated version addresses all of the concerns that were raised during its previous iteration. The authors have done a great job at addressing the concerns and incorporating the suggestions in this updated version.

Reviewer #3: I applaud the authors for carrying out a well-designed study on the suppressive effects of midazolam versus S-ketamine on respiratory rate and tidal volume variability. The results suggest that S-ketamine may be a safer option for preserving the respiratory system's adaptability at a dose that is not sufficient to depress the respiratory rate.

There are several inherent limitations of the study:

A. The non-invasive bioimpedance technique (ExSpiron® Xi) has been used to measure respiratory rate (RR) and tidal volume (TV) variation; however, it is not widely accepted for measuring respiratory parameters in clinical practice. It is also uncertain whether these study findings can be replicated in future studies or clinical practice.

B. Although authors used a linear mixed model for statistical analysis, the 22% missing data still appears substantial, primarily due to missing baseline and intervention data resulting from poor capture and inadequate storage. This highlights a key limitation of the non-invasive bioimpedance technique used in this trial.

C. Although the dosing regimen used in this trial matches a previous study of fibromyalgia patients, it may still differ significantly in real-world clinical practice, raising concerns about the generalizability of this data to the ICU population. Additionally, during paired spontaneous awakening and breathing trials before extubation in the ICU, all sedatives are typically stopped, which makes the authors' claim that midazolam poses a higher risk of extubation failure compared to S-ketamine an overstretch of the study’s findings. Interestingly, the dose used in this study didn’t significantly lower the respiratory rate. Still, the variation in respiratory and tidal volume was found to be significant, suggesting impaired adaptability of the respiratory system, which may still have clinical implications.

However, I believe the authors have addressed all these limitations in the article. Building on the constructive feedback from the previous reviewers and the authors’ thorough revision, the article has seen notable improvements in both content and layout. The revised introduction, methodology, and discussion sections now more clearly highlight the study’s novelty, scope, design, and findings, making the article more engaging and informative for its audience. These enhancements demonstrate a positive progression and better alignment with current publication standards.

Reviewer #4: No comments provided since I believe the manuscript is now acceptable for publication, thank you very much.

Reviewer #6: I am concerned about the limited sample without appropriate exclusion criteria, not including side-effects of medications, C02 levels, and quality of sedation, and missing data.

7. PLOS authors have the option to publish the peer review history of their article (what does this mean?). If published, this will include your full peer review and any attached files.

Reviewer #1: **Yes: **Sathish Kumar Krishnan

Reviewer #2: **Yes: **Dili Dhanani MD

Reviewer #3: **Yes: **Abhisekh Sinha Ray

Reviewer #4: No

Reviewer #6: No

---

## [Editor Report · Acceptance letter]

PONE-D-25-20310R1

PLOS ONE

Dear Dr. van den Bosch,

I'm pleased to inform you that your manuscript has been deemed suitable for publication in PLOS ONE. Congratulations! Your manuscript is now being handed over to our production team.

Kind regards,

on behalf of

Dr. Kartikeya Rajdev

Academic Editor

PLOS ONE